

# Role of Pink1-mediated mitophagy in adenomyosis

Minmin Chen[1,2,3,*], Wei Wang[1,*], Xianyun Fu[1,2,3], Yongli Yi[2,3], Kun Wang[1,2] and Meiling Wang[1,2]

[1] College of Traditional Chinese Medicine, China Three Gorges University & Yichang Hospital of Traditional Chinese Medicine, Yichang, China
[2] Third-grade Pharmacological Laboratory on Traditional Chinese Medicine, State Administration of Traditional Chinese Medicine, China Three Gorges University, Yichang, China
[3] College of Medicine and Health Sciences, China Three Gorges University, Yichang, China
[*] These authors contributed equally to this work.

## ABSTRACT

**Abstract Background**. Recent studies indicate that endometrial hypoxia plays a critical role in adenomyosis (AM) development. Mitochondria are extremely sensitive to hypoxic damage, which can result in both morphological and functional impairment. Mitophagy is a crucial mechanism for preserving mitochondrial quality by selectively removing damaged mitochondria, thus ensuring the proper functioning of the entire mitochondrial network. In response to hypoxia, PINK1 is activated as a regulator of mitophagy, but its role in AM requires further study.

**Objective**. To explore the potential mechanism of mitophagy mediated by PINK1 in the pathogenesis of AM.

**Method**. The study compared PINK1, Parkin, OPTIN, P62, and NDP52 protein expression levels in patients with or without AM using clinical specimens and an AM mouse model. Pathological changes were compared using HE staining. Immunofluorescence and western blot were used to detect protein expression levels. Endometrial stromal cells (ESC) were isolated and examined for mitophagy, protein expression level, and cell invasion ability.

**Results**. Both the endometrial tissue from patients with AM and AM ESC displayed an upregulation of protein levels for PINK1, Parkin, OPTIN, P62, and NDP52 when compared with the control group. Then, HE staining confirmed the successful establishment of the AM mouse model. Moreover, the ultrastructural analysis using transmission electron microscopy revealed that AM mice's endometrial glandular epithelial and stromal cells had exhibited swollen, deformed, and reduced mitochondria along with an increase in the number of lysosomes and mitochondrial autophagosomes. The protein levels of PINK1, Parkin, OPTIN, P62, and NDP52 in uterine tissue from AM mice were noticeably increased, accompanied by a considerable upregulation of ROS levels compared to the control group. In addition, cells in the AM group showed remarkably elevated mitophagy and invasion potentials compared to the control group. In contrast, the cell invasion ability decreased following PINK1 knockdown using the RNA interference technique.

**Conclusion**. The high levels of PINK1-mediated mitophagy have been found in AM. The upregulation in mitophagy contributes to mitochondrial damage, which may result in the abnormal invasion characteristic of AM.

Corresponding authors
Xianyun Fu, dinnar1@163.com
Yongli Yi, 728113811@qq.com

## INTRODUCTION

Adenomyosis (AM) is a common condition among women of reproductive age, characterized by an increased menstrual volume, progressive dysmenorrhea, and chronic pelvic pain. AM is known to be responsible for approximately 10% of subfertility cases (*Mishra et al., 2023*). The recurrence of symptoms, clinical resistance, and the unfavorable fertility prognosis have a tremendous negative impact on the physiology and psychology of patients with AM (*Alcalde et al., 2021*).

AM is generally considered to be a benign disease. However, it shares some characteristics with malignant tumors, such as an invasive growth of the endometrium into the myometrium. Multiple pathogenic factors influence the development of AM, and its progression is closely related to the complex microenvironment in the endometrial-myometrial interface (*Lazaridis et al., 2022*). Although there have been numerous attempts to explore the underlying mechanisms and develop drugs that target immune abnormalities, endocrine disorders, and oxidative stress, clinical challenges associated with AM remain unresolved (*Moldassarina, 2023*). Therefore, it is imperative to develop a novel perspective in studying AM.

Our previous studies suggest that hypoxia is one of the vital triggering mechanisms that initiate the development of AM (*Fu Xianyun, 2015*). Mitochondria are the most advanced intracellular reactive sensing systems, susceptible to hypoxic damage. Generally, when the mitochondrial network is affected by hypoxia, the damaged mitochondria will be selectively removed through mitophagy to regulate mitochondrial mass (*Sun & Airong, 2010*). Recent research has identified PTEN-induced putative kinase 1 (PINK1) as the primary detector of mitochondrial damage. PINK1 prompts the translocation of Parkin to the outer mitochondrial membrane, where it activates ubiquitin ligase function. Autophagy receptor proteins, such as P62, NDP52, OPTIN, and others, can recognize the ubiquitinated mitochondria and bind to the membrane of autophagic vesicles (*Wang et al., 2023b*). These vesicles will fuse with lysosomes to form mito-autolysosomes that remove damaged mitochondria. Mitophagy plays a vital role in regulating mitochondrial quality control. Disrupted mitophagy has been shown to cause abnormal cell proliferation (*Larson-Casey et al., 2016*; *Shida et al., 2016*). However, the involvement of mitophagy in AM remains poorly studied.

Therefore, this study aims to explore the potential mechanism of AM occurrence by targeting the changes in mitophagy through *in vivo* and *in vitro* experiments. The results from this research will offer a novel perspective for the investigation of AM.

## MATERIALS AND METHODS

### Materials

#### Experimental animals

Six female and two male 7-week-old ICR mice weighing $25 \pm 3$ g were purchased from Beijing Weitong Lihua Laboratory Animal Technology Co., Ltd. The mice were maintained under specific pathogen-free (SPF) conditions.They were provided with unlimited access to food and water at a temperature of $22 \pm 2\,°C$, relative humidity of $50 \pm 5\%$, and a 12-hour light-dark cycle. The animal experiment and welfare strictly adhered to the relevant guidelines, and the experiment procedures were approved by the Animal Experiments Ethics Committee of the Medical College of China Three Gorges University (Approval No. 2019080I).

#### Drugs and reagents

The following antibodies were purchased from Wuhan Sanying Biotechnology Co., Ltd: Rabbit polyclonal antibodies against PTEN-induced putative kinase 1 (PINK1) (batch number 23274-1-AP), Parkinson protein 2 (Parkin 2) (batch number 14060-1-AP), optineurin (OPTIN) (batch number 10837-1-AP), and nuclear dot protein 52 (NDP52) (batch number 12229-1-AP). The rabbit polyclonal antibody against glyceraldehyde-3-phosphate dehydrogenase (GAPDH) (batch number AB-P-R001) was purchased from Hangzhou Xianzhi Biological Technology Co., Ltd. Goat anti-rabbit IgG HRP conjugate (batch number BA1054) was purchased from Wuhan Boshide Biological Engineering Co., Ltd. The BCA protein concentration determination kit (batch number P0010), Mito-Tracker Green mitochondrial probe (batch number 040721210826), Lyso-Tracker Red lysosomal probe (batch number 012621210827), and ROS detection kit (batch number S0033) were all purchased from Beyotime Biotechnology Co., Ltd. Propofol emulsion injection (National Drug Approval No. H19990282, batch number 21803281) was purchased from Xi'an Libang Pharmaceutical Co., Ltd. Lipofectamine®2000 (batch number 2366512) was purchased from Invitrogen Corporation in the United States. PINK1 siRNA (batch number PV211123002) was purchased from Wuhan Paivai Biotechnology Co., Ltd.

#### Instruments

MCO175 CO2 incubator (Sanyo, Osaka, Japan); Multiskan GO full wavelength microplate reader (Thermo Fisher Scientific, Waltham, MA, USA); electrophoresis apparatus, 170-8280 ChemiDoc MP chemiluminescence imaging system (Bio-Rad, Hercules, CA, USA); BOM-1000 ultra-clean workbench (Suzhou Antai Airtech Co., Ltd., Suzhou, China); 2391 Model transmission electron microscope (TEM) (IITC, USA).

### Research methods

#### Sample collection

Six uterine samples were collected from patients who underwent hysterectomies at the Three Gorges University affiliated First People's Hospital from September 2021 to July 2022. A total of six samples were included in this study. The model group consisted of three samples from patients with AM, confirmed by pathological biopsy. The remaining three

samples were obtained from patients with uterine leiomyoma and served as the control group. We have explained in the methods section that we have received written informed consent from participants of our study. This study was approved by the Medical Ethics Committee of China Three Gorges University (Approval No. 2019CA73). Consent forms were obtained from all participants involved in the study to ensure their informed consent.

### Animal experiments

*(1) Establishment and grouping of animal models.* According to the 4R principles for animal research, the sample size was determined to be six mice per group to obtain statistical power. AM is routinely experimentally induced in mice through neonatal tamoxifen exposure (*Marquardt, Jeong & Fazleabas, 2020*). With a mating ratio of 3 females to 1 male, 12 female neonatal mice will be obtained. The neonatal mice were randomly divided into the control group ($n = 6$) and the model group ($n = 6$). Microsoft Excel generated random sequences. From postnatal day 2 to 5, the mice of the model group were gavaged with 5 $\mu$ l/g of a mixture containing peanut oil, lecithin, and condensed milk (2:0.2:3 volume ratio), along with 2.7 $\mu$ mol/kg tamoxifen citrate. The control group was given an equal volume of saline. The mice in each group were appropriately labeled, and each procedure was performed sequentially according to mouse numbering order to control the confounders. All the mice were euthanized at 8 weeks by cervical dislocation after inhalational anesthesia. The uterine tissues of mice were collected, and all samples were stored at $-80\,°C$.

*(2) Hematoxylin and eosin (HE) staining.* The uteri of each group were subjected to fixation with 4% paraformaldehyde, followed by embedding in paraffin. The tissues were allowed to dry at 65 °C for 30 min and were then cut into 5 $\mu$m thick serial sections. After staining with HE, pathological infiltration was observed under a microscope.

*(3) The ultrastructure of endometrial epithelial cells and stromal cells.* The isolated tissues, with a thickness of one mm, were subjected to double fixation with glutaraldehyde and osmic acid. Then, the tissues were dehydrated using an acetone gradient and treated with propylene oxide before embedding and polymerization. Ultra-thin slices were obtained using a microtome and positioned with toluidine blue. Finally, the pieces were stained sequentially with uranyl acetate and lead citrate to observe the presence of mitochondria, lysosomes, and mitophagosomes.

*(4) Reactive oxygen species (ROS) levels were detected.* The tissues were solidified with OCT embedding agent, and then sliced to a thickness of 8 um. The sections were incubated by DHE (100uM) solution at 37 °C for 30 min. After being washed with PBS for three times, the DAPI staining solution was added. After incubation at room temperature for 10 min, away from light, the sections were then washed three times again. Finally, an anti-fluorescence quencher was added, and the sealed sections were observed under a fluorescence microscope.

*(5) The localization and quantification of PINK1 and Parkin.* The uterus sections were initially exfoliated and rinsed before adding goat serum, followed by PINK1 antibody, Parkin antibody, and secondary antibody consecutively. The sections were then rinsed thrice with PBS solution for 5 min each time. T DAPI was introduced into the sections, after which they were incubated in PBS solution thrice for 5 min each to counterstain the nuclei. Subsequently, anti-fluorescence quencher was added to the sections before sealant application. Finally, the staining results were observed under a fluorescence microscope.

*(6) The expression of mitophagy protein.* The BCA protein quantification kit was utilized to determine the protein concentration of each group. Next, protein samples were loaded onto a 10% sodium dodecyl sulfate-polyacrylamide gel for electrophoresis and transferred to a PVDF membrane. The membrane was blocked at room temperature, followed by rinsing and the addition of antibodies against NDP52, OPTIN, PINK1, Parkin, and GAPDH (1:1000), respectively. The samples were incubated at 4 °C overnight, and after washing the membranes, HRP-labeled secondary antibody (1:10000) was added. Lastly, the membranes were incubated at room temperature for 2 h, rinsed, and exposed to chemiluminescent reagents. The value of the protein bands was measured using Image-Pro Plus software.

### In vitro experiments

*(1) Isolation and culture of primary endometrial stromal cells (ESCs).* The freshly isolated endometrial tissues were immediately placed in a phosphate buffer solution (PBS) and transported to the laboratory within 30 min for further processing. The intimal tissue was cut into small pieces of approximately one mm and incubated with 3-5ml of type I collagenase solution (with a concentration of 2mg/mL) after washing with PBS. After mixing, the tissue was digested intermittently at 37 °C with shaking for 40-60 min. The tissue was first filtered through a 100-mesh copper mesh and then through a 400-mesh filter. The lower filtrate containing interstitial cells was rinsed with a culture medium and centrifuged at 1000 rpm for 15 min. The resultant stromal cells were washed, seeded onto 6-well plates with culture medium (covered with coverslips), and incubated at 37 °C in a 5% $CO_2$ humidified atmosphere. The purity of the cell samples was identified through vimentin and cytokeratin staining.

*(2) Identification of AM ESCs.* The freshly isolated endometrial tissues were immediately placed in AM ESCs, digested and separated into a single-cell suspension, and seeded into Petri dishes containing coverslips. Once the cells had adhered to the slips, they were fixed using 4% paraformaldehyde and washed with PBS. Next, the cells were permeabilized using 0.5% Triton-X-100 at room temperature for 20 min before being immersed in PBS. The slides were blocked by dropping 3%BSA at room temperature for 30 min. The diluted primary antibody (vimentin (1:200)) was added dropfold, and the reaction was performed overnight at 4 °C. After washing the cell slides with PBS, HRP-labeled secondary antibody was added. After washing with PBS, a diluted CY3-TSA working solution was added, and the cells were incubated for 10 min in the dark. DAPI was added to restain cell nuclei for 10 min at room temperature; the slide was sealed with an anti-fluorescence quenching sealing agent and photos were taken under a fluorescence microscope.

*(3) Knockdown of PINK1.*  To knock down the PINK1 gene, ESCs were seeded in 6-well plates of $5 \times 10^5$ cells/well density during their logarithmic growth phase. The cells were then cultured overnight in DMEM medium at 37 °C in a carbon dioxide incubator. Following the instructions of Lipofectamine TM3000 transfection reagent, si-PINK1 and its negative control (scramble) were transfected into the cells. After 6 h of transfection, the cells were cultured with a complete medium until 48 h. Transfection efficiency was evaluated using immunofluorescence and Western Blot techniques to determine the optimal sequence of si-RAGE.

*(4) Assessment of invasion abilities.*  In the Transwell Invasion assay, 100 μL of diluted Matrigel was added to each chamber and incubated in a 37 °C incubator for 4-5 h. The Matrigel was prepared by dilution in a serum-free medium at a ratio of 1:8. Afterwards, AM ESC concentration was adjusted to $5 \times 10^5$ cells/mL using a serum-free medium. Next, 200 $\mu$L of the cell solution was placed into the upper chamber of the Transwell. Then, 800 μL of complete culture medium containing 10% fetal bovine serum was added to the lower chamber, and the chambers were incubated for 24 h at 37 °C and 5% $CO_2$. The migrated cells were fixed using methanol for 20 min and stained with 0.1% crystal violet for 20 min. The chambers were subsequently rinsed in running water to remove any excess stains. Finally, the chambers were observed under a microscope and photographed. This procedure was repeated three times to ensure the accuracy and reliability of the results.

*(5) The level of mitophagy in the ESCs.*  Two fluorescent probes were prepared: Mito-Tracker Green, a mitochondrial-specific probe, and Lyso-Tracker Red, a lysosome-specific probe. These probes were then incubated at 37 °C for 40 min. Subsequently, the staining solution was extracted, the cell culture dish was replenished with fresh medium, and the cells were observed under a laser confocal microscope. In the resulting images, the mitochondria were depicted as green fluorescent spots while lysosomes appeared as red fluorescent spots. Cells displaying mitophagy demonstrate a co-localization of yellow fluorescence emanating from overlapping the green and red signals. Therefore, these yellow spots are deemed to be mitophagy-positive.

*(6) The expression of mitophagy proteins.*  The process following cell protein extraction is identical to step 2.2.2 (6).

### Statistical analysis

Fluorescence images were analyzed using image-Pro Plus software. Statistical analyses were performed with SPSS 25.0 (IBM Corp, Armonk, NY, USA). Data were tested for normality using the SPSS normality test. Results are expressed as mean $\pm$ SD. *$P <$ 0.05, **$P <$ 0.01, and **$P <$ 0.001 were considered statistically significant. To ensure the objectivity of the statistical analysis, after the allocation and the conduct of the experiment, the outcome assessment and the data analysis were completed independently by the experimentalists, who were unaware of the group allocation.

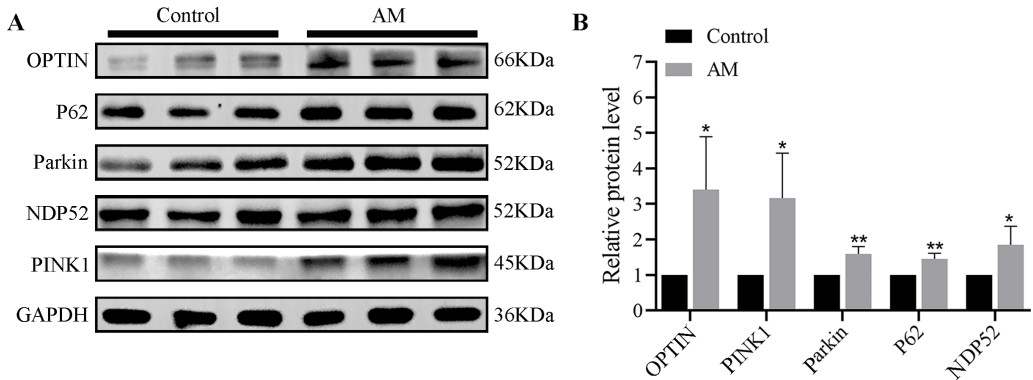

**Figure 1 (A–B) Protein expression of PINK1/Parkin mitophagy pathway ($n = 3$ per group).** Compared with the control group, $*P < 0.05$, $**P < 0.01$.

# EXPERIMENTAL RESULTS

## Differences in protein expression levels of PINK1/Parkin mitophagy pathway derived from human utopic endometrium tissues

The Western blot findings indicated a significant increase in the expressions of mitophagy-related proteins such as PINK1, Parkin, OPTIN, P62, and NDP52 in the AM group compared to the control group (Fig. 1).

## Results of animal experiments

### The pathological changes

Figures 2A and 2B demonstrate a distinct boundary between endometrial and muscular layers in the control group. However, following the modeling process, endometrial glandular epithelial cells and stromal cells infiltrated in the myometrium, as observed in Figs. 2C and 2D.

### Ultrastructural changes

Mitophagy in eutopic endometrium was observed using transmission electron microscopy. The study found that the control group had abundant mitochondria with normal size and morphology, and clear mitochondrial ridges in both stromal and glandular epithelial cells (stromal cells: Figs. 3A–3B; glandular epithelial cells: Figs. 3E–3F). However, in the AM group, the cells' mitochondria were enlarged and some showed vacuolar degeneration. The typical ridge structure disappeared, and there was an increase in the number of surrounding lysosomes. Some mitochondria merged with lysosomes to form mitophagosomes characterized by double-membrane structure (stromal cells: Figs. 3E–3F; glandular epithelial cells: Figs. 3G–3H).

### The localization and expression of PINK1 and Parkin

This study utilized the immunofluorescence staining technique to investigate the localization and expression alteration of PINK1 and Parkin proteins in both control and AM groups. Our findings in Figs. 4A and 4B, highlight the expression of PINK1 and Parkin proteins in various cell types, including glandular epithelial cells, stromal cells,
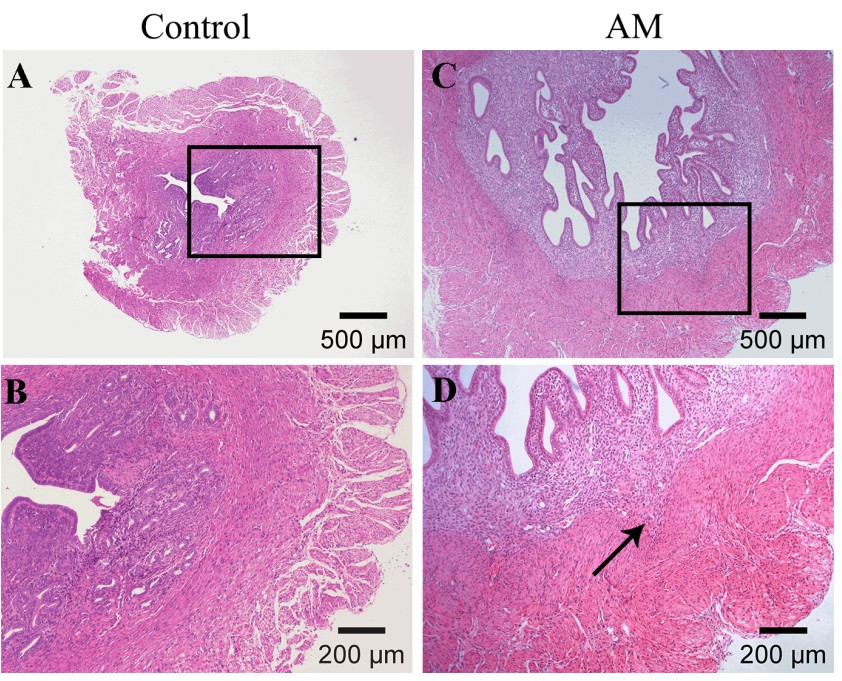

Control          AM

**Figure 2** **HE staining of mouse uterine tissue transverse sections.** (A–B) Control group; (C–D) model group. Arrows indicate invasion of endometrial stromal cells and glandular epithelial cells into the myometrium.

and smooth muscle cells in both groups. Notably, our results demonstrate a significant upregulation in the expression of PINK1 and Parkin proteins within the endometrial layer of the AM group when compared to the control group.

### Alteration in ROS levels

Compared to the control group, higher level of ROS was observed in the eutopic endometrium of mice in the AM group (Fig. 4C), as evidenced by the findings from fluorescent probe labeling.

### Protein expression levels of PINK1/Parkin mediated mitophagy pathway

Compared with the control group, the expression levels of mitophagy-related proteins, including PINK1, Parkin, OPTIN, P62, and NDP52, were found to be significantly elevated in the AM group (Fig. 5), as evidenced by the Western blot analysis.

## Results of *in vitro* experiments
### The level of mitophagy

Immunofluorescence double labeling was used to observe the changes in mitophagy between the control group and the AM group. Figures 6A–6C show a significant increase in the number of co-localizations between mitochondria and lysosomes in the AM group compared to the control group.

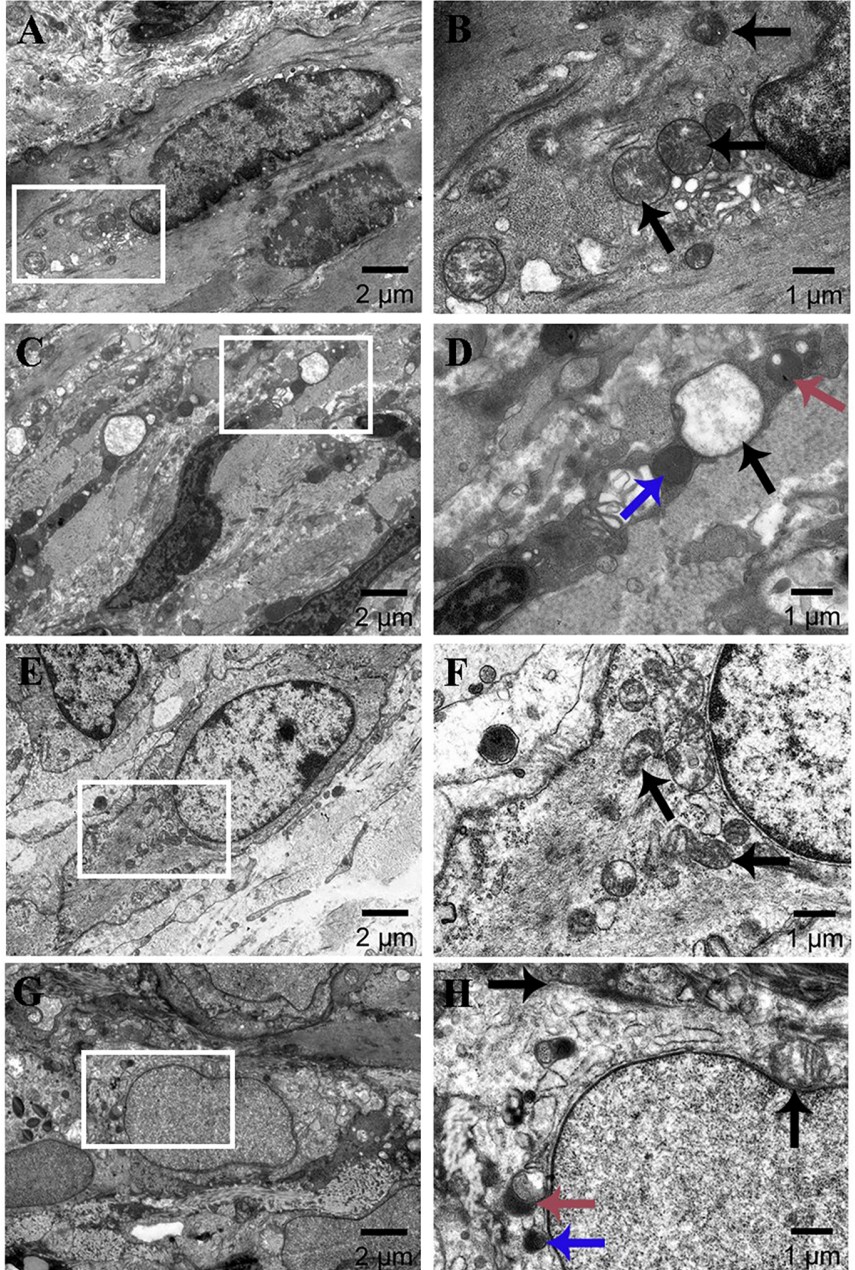

**Figure 3 Ultrastructural changes of eutopic endometrium (×2k, ×6k).** (A–B) Stromal cells in the control group; (C–D) stromal cells in the AM group; (E–F) glandular epithelial cells in the control group; (G–H) glandular epithelial cells in the AM group. Black, blue, and red arrows indicate mitochondria, lysosomes, and mitophagosomes.

### Protein expression of PINK1/Parkin mediated mitophagy pathway

The Western blot results suggest a significant increase in the levels of the proteins associated with mitophagy, including PINK1, Parkin, OPTIN, P62, and NDP52 in the AM group compared to the control group (Fig. 6D).

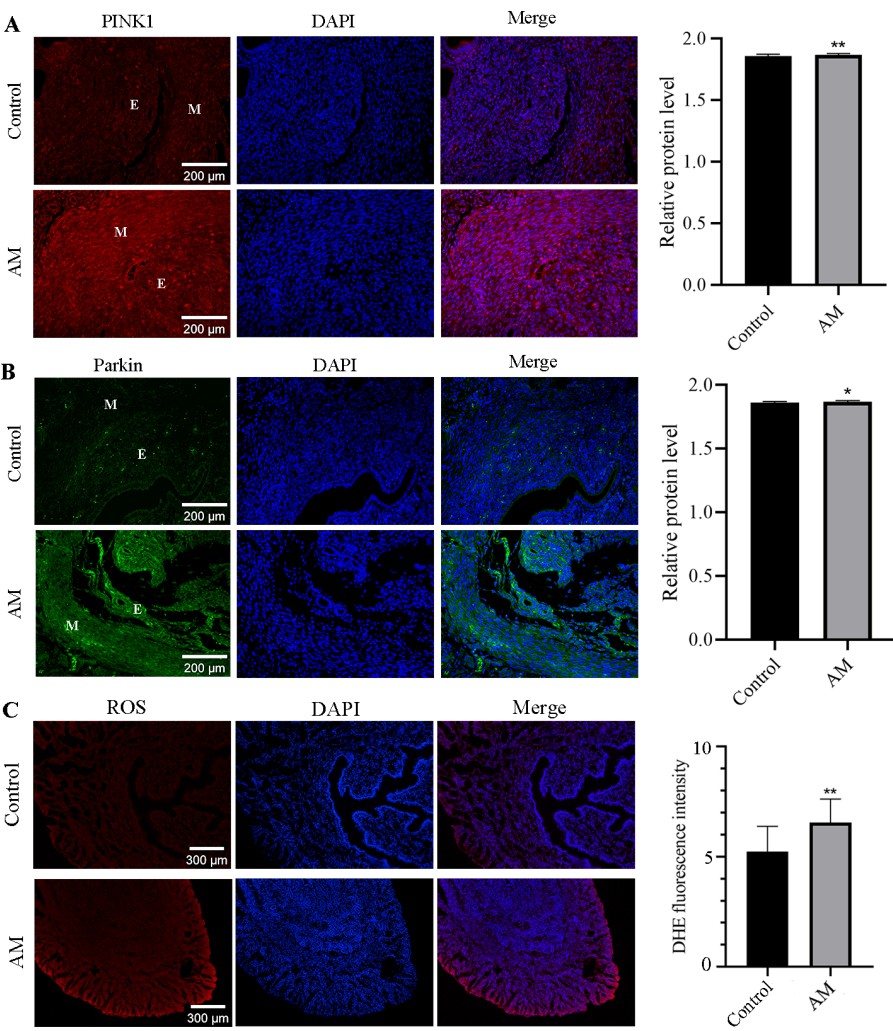

**Figure 4** **Immunofluorescence Staining and ROS.** (A and B) Localization and expression of PINK1 and Parkin proteins in mouse uterine tissues. (M: uterine smooth muscle layer; E: endometrial layer.) (C) ROS levels in eutopic endometrium. Compared with the control group, $*P < 0.05$, $**P < 0.01$.

### Protein expression of PINK1 after transfection

A Western blot experiment examined the expression of the PINK1 gene in cells of each group. The results revealed a reduction in the expression levels of PINK1 protein in the si-1, si-2, and si-3 groups compared to the AM group. Notably, among them, the impact si-3 in downregulating the expression of PINK1 was more prominent, with a statistically significant difference ($P < 0.05$, Fig. 7A). As a result, si-3 was deemed suitable for subsequent experiments. These findings suggest that siRNA technology can effectively manipulate the expression of the PINK1 gene in cells.

### Transwell invasion assay

The Transwell invasion assay revealed that the AM group had a significantly higher number of ESCs than the Control group, suggesting that AM ESCs possess strong invasive properties

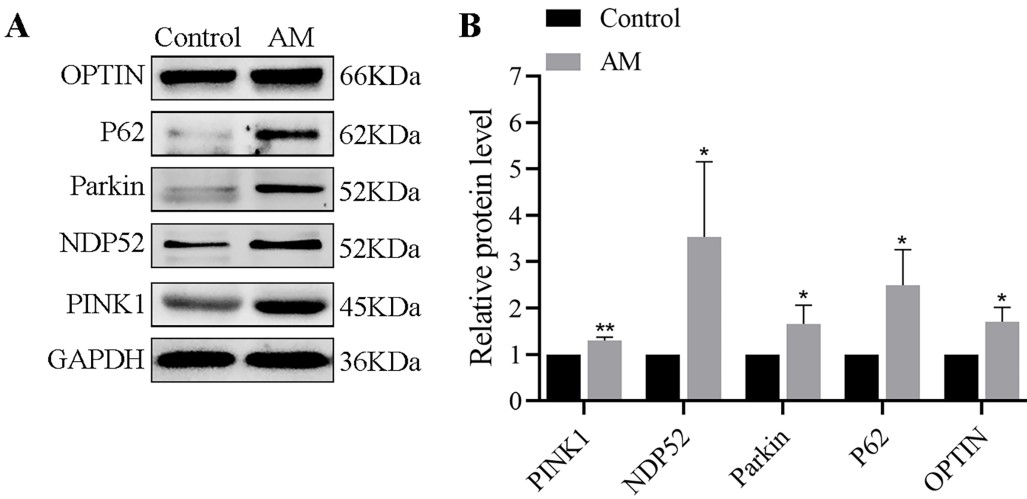

**Figure 5** (A–B) Protein expression of PINK1/Parkin-mediated mitophagy pathway detected by Western blot in mouse uterine tissues ($n = 6$ per group). Compared with the control group, $*P < 0.05$, $**P < 0.01$; ns, no significant.

(Fig. 7B). Using siRNA technology to knock down PINK1 expression demonstrated that the number of cell invasions in the PINK1-siRNA group decreased significantly compared to that in the AM group (Fig. 7C). This finding indicates that the inhibition of the expression of PINK1 resulted in a significant reduction in the invasion ability of AM ESCs.

## DISCUSSION

Adenomyosis (AM) is a widespread medical condition that affects a considerable proportion of women, leading to symptoms including menorrhagia (50% of cases), dysmenorrhea (30% of cases), uterine bleeding (20% of cases), and infertility (30% of cases) (*Harada et al., 2016*). These manifestations can critically impact women's quality of life, causing both physical and emotional discomfort. The pathological features of AM are characterized by the invasive growth of the endometrium into the myometrium, exhibiting malignant proliferation and metastasis similar to tumors. AM is not caused by a single pathogenic factor. Instead, the endometrial-myometrial Interface (EMI) exists in a highly complicated microenvironment where multiple signaling pathways interact. The effectiveness of current conservative treatments targeting endocrine or inflammatory immunity is limited due to the tolerability and safety (*Suardika & Astawa, 2018*). Therefore, it is imperative to discover new mechanisms underlying the disease and potential therapeutic targets.

The natural history of AM lesions is undergoing repeated tissue injury and repair, accompanied by increased endometrial fibrosis, contributing to hypoxia (*Guo, 2020*). Mitochondria are an advanced system that senses and responds to intracellular reactive stimuli, especially sensitive to hypoxic damage. When the mitochondrial network is damaged due to a hypoxic environment, it triggers a process known as 'selective autophagy'. We tested mitophagy-related proteins in clinical uterine tissue specimens and confirmed that they were significantly upregulated in AM, as shown in Fig. 1. Subsequently,

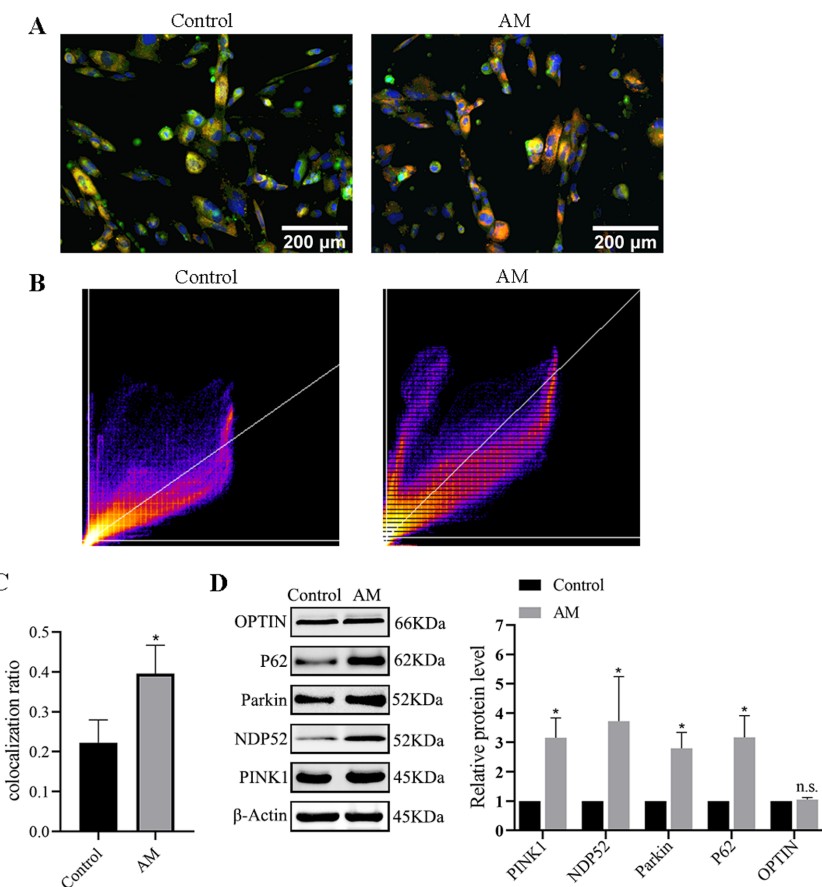

**Figure 6 Mitophagy levels in cells.** (A) Fluorescence images and co-localization diagrams of mitochondria and lysosomes. (B) Pearson correlation coefficient analysis graph. (C) Colocalization rate = colocalized cells/total cell count. (D) Protein expression levels of the PINK1/Parkin-mediated mitophagy pathway. Compared with the control group, $*P < 0.05$, $**P < 0.01$; ns, no significant.

we conducted experiments both *in vivo* and *in vitro* to further investigate the specific mechanism.

In contrast to the pituitary engraftment model, the tamoxifen treatment method for inducing AM is more easily defined and controlled at the molecular level, and it does not require surgical manipulation that may affect lesion development (*Marquardt, Jeong & Fazleabas, 2020*). Previous experiments of our research have successfully established a stable mouse AM model by neonatal dosing with tamoxifen (*Liao and ZJ , 2020*), and pathological HE staining in this study also confirmed the success of AM modeling, as shown in Fig. 2. Ultrastructural observations in Fig. 3 revealed that the number of mitochondria in the endometrium of control mice was high, with normal morphology and visible mitochondrial bilayer membranes and ridges. In the AM model mice, the number of mitochondria was significantly reduced in both uterine stromal cells and glandular epithelial cells, as well as swollen mitochondria and vanished mitochondrial ridges.

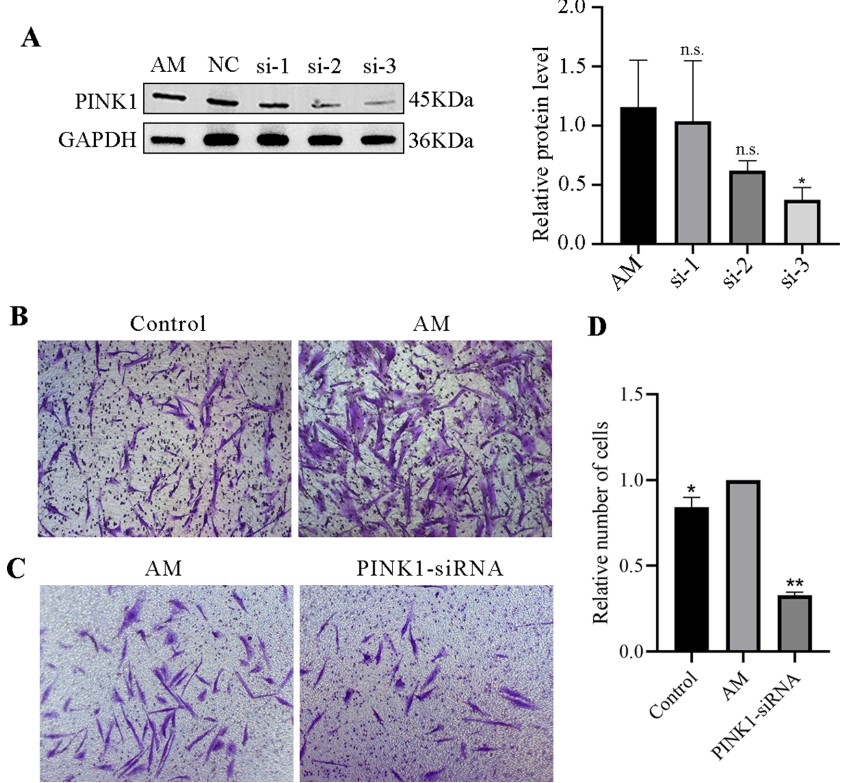

**Figure 7** **PINK1-siRNA and Transwell invasion assay.** (A) Western blot was performed to detect the expression of PINK1 protein in cells from each group, compared with the AM group, $*P < 0.05$; ns, no significant. (B–D) Transwell invasion assay. Compared with the AM group, $*P < 0.05$, $**P < 0.01$.

The most common intracellular source of ROS is mitochondrial metabolism (*Wang et al., 2023a*). Mitochondrial dysfunction can trigger oxidative stress due to the loss of electrons during the transfer process in the respiratory chain, generating reactive oxygen species (ROS). Oxidative stress has been demonstrated to be closely related to the severity of endometriosis, with ROS products serving as prognostic indicators for endometriosis patients (*Sun & Airong, 2010*). Researchers have found that the level of ROS in cancer cells is a double-edged sword. Moderately high levels of ROS are beneficial to maintain tumor cell genesis and development, while toxic levels of ROS have been shown to be an important force in destroying cancer cells (*Zhao et al., 2023*). Our research has verified that the pathological progression of endometrial invasion into the myometrium is accompanied by the excess accumulation of ROS, as shown in Fig. 4C. This may ultimately contribute to the the abnormal invasion ability. However, the specific mechanism needs to be further studied.

The defective mitochondria are selectively cleared to maintain overall mitochondrial mass (*Khan et al., 2015*). Autophagic vesicles and their association with damaged mitochondria play critical roles in this cellular process (*Yoo & Jung, 2018*). Recent studies have highlighted the importance of the PINK1 and parkin signaling pathways in regulating

mitophagy. PINK1 significantly accumulates in mitochondria with reduced potential and is the primary detector of mitochondrial damage. This serine/threonine protein kinase is typically present in low levels in healthy mitochondria (*Thomas et al., 2014*). However, when the mitochondria become depolarized, the degradation pathway of PINK1 is disrupted, and its autophosphorylation is triggered in the outer mitochondrial membrane (OMM), leading to a buildup of phosphorylated PINK1. As a result, PINK1 is used as a marker for damaged mitochondria. Activated PINK1 primarily recruits Parkin to OMM of damaged mitochondria. Parkin functions as a ubiquitin-protein ligase enzyme (E3) (*Harper, Ordureau & Heo, 2018*), which forms phosphorylated polyubiquitin chains at the OMM and mediates the polyubiquitination of a wide range of OMM proteins. It is one of the most widely studied E3 ubiquitin ligases. PINK1 and parkin constitute a feedforward mechanism of mitophagy (*Bonam, Ruff & Muller, 2019*). PINK1 plays a crucial role in facilitating the translocation of parkin from the cytosol to the damaged outer mitochondrial membrane, leading to the activation of ubiquitin ligase activity. Autophagy receptor proteins, such as P62, NDP52, OPTIN, and others, can precisely identify ubiquitinated mitochondria, bind to the autophagic vesicle membrane, and subsequently merge with lysosomes to form mito-autolysosomes, which ultimately eliminate impaired mitochondria. Therefore, mitophagy serves as a critical component of mitochondrial quality control.

Currently, the regulation of mitophagy in endometriosis is controversial. Some researchers suggest that down-regulating mitophagy can significantly reduce the endometrial stromal cells' migration and survival ability (*Zhao et al., 2018*). In contrast, others argue that activating the mitophagy pathway reduces lesion volume, area, and diameter (*Siracusa et al., 2021*). In this study, Fig. 5 present that the uterine specimens from the AM mice displayed an up-regulation of PINK1, Parkin proteins, as did their downstream proteins: OPTIN and NDP52. We also observed a significant increase in p62 levels in AM. Although increased p62 levels are generally considered a marker of autophagy dysfunction, multiple studies have also demonstrated that an elevation in p62 levels is often paralleled by activation of autophagy under oxidative stress. This may be attributed to the p62, serving as a stress protein, experiencing significant upregulation under stressful conditions. The observation through transmission electron microscopy also confirmed a marked elevation in the number of lysosomes surrounding the defected mitochondria, alongside a corresponding rise in mitophagosomes. These results indicate that the over-activated mitophagy in AM leads to the removal of damaged mitochondria rather than the repair of damaged mitochondria, reducing mitochondrial number and eventually leading to abnormal and defective cell function.

Similarly, *in vitro* experimentation disclosed increased colocalization of mitochondria and lysosomes in AM ESCs, accompanied by intensified mitophagy and increased expression of PINK1-related mitophagy pathway proteins, as shown in Fig. 6. The method of knocking down PINK1 was adapt to verify the effect of mitophagy on AMESC cells. The optimal PINK1-siRNA sequence were selected from among the three candidate sequences shown in Table 1. Figure 7 demonstrates that after PINK1 knockdown, the invasion ability of AM ESCs was significantly reduced, indicating that abnormal mitophagy may contribute to the formation of AM-associated invasion ability.

**Table 1** The synthesis of siRNA.

| Name | Sequence (5′→ 3′) |
| --- | --- |
| hPINK1 si-1 sense | CGCUGUUCCUCGUUAUGAATT |
| hPINK1 si-1 antisense | UUCAUAACGAGGAACAGCGTT |
| hPINK1 si-2 sense | AUGGGUCAGCACGUUCAGUUATT |
| hPINK1 si-2 antisense | UAACUGAACGUGCUGACCCAUTT |
| hPINK1 si-3 sense | GAAAUCCGACAACAUCCUUTT |
| hPINK1 si-3 antisense | AAGGAUGUUGUCGGAUUUCTT |

In summary, our findings suggest that PINK-mediated mitophagy is abnormally activated in AM, resulting in the excessive removal of hypoxia-damaged mitochondria, ultimately leading to mitochondrial dysfunction and homeostasis disruption. The insights from our study provide a valuable contribution towards identifying the underlying pathogenic mechanisms and potential therapeutic targets for AM. Additionally, our findings underscore mitophagy's critical role in disease conditions, emphasizing the need to investigate further its involvement in various health disorders linked with chronic hypoxia.

This study has a few potential limitations that need to be acknowledged. Firstly, the sample size employed in this research is relatively small, which may result in reduced statistical power. Secondly, since the study exclusively utilized a mouse model, it may impose certain limitations on the extent of exploring the disease process.

### Funding

The funding for this research was provided by the National Natural Science Foundation of China (No.81973897). The funders had no role in study design, data collection and analysis, decision to publish, or preparation of the manuscript.

### Grant Disclosures

The following grant information was disclosed by the authors:
National Natural Science Foundation of China: 81973897.

### Competing Interests

The authors declare there are no competing interests.

### Author Contributions

- Minmin Chen performed the experiments, analyzed the data, prepared figures and/or tables, and approved the final draft.
- Wei Wang conceived and designed the experiments, authored or reviewed drafts of the article, and approved the final draft.
- Xianyun Fu conceived and designed the experiments, authored or reviewed drafts of the article, and approved the final draft.

- Yongli Yi performed the experiments, prepared figures and/or tables, and approved the final draft.
- Kun Wang analyzed the data, authored or reviewed drafts of the article, and approved the final draft.
- Meiling Wang analyzed the data, prepared figures and/or tables, and approved the final draft.

## Human Ethics

The following information was supplied relating to ethical approvals (i.e., approving body and any reference numbers):

Medical Ethics Committee of China Three Gorges University

## Animal Ethics

The following information was supplied relating to ethical approvals (i.e., approving body and any reference numbers):

Approval for the experimental protocol was granted by the Medical Animal Ethics Commission of China Three Gorges University, under the reference number NO. 2019080I.

## Data Availability

The original bands of Western blot are available in the Supplementary File.

## Supplemental Information

Supplemental information for this article can be found online at http://dx.doi.org/10.7717/peerj.16497#supplemental-information.

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
