# Peer review of "Role of Pink1-mediated mitophagy in adenomyosis"

_PeerJ, doi:10.7717/peerj.16497_

## Round 0.1 · original submission · Major Revisions

Please address the concerns of all reviewers and amend the manuscript accordingly.

Reviewer 1 ·

Basic reporting

• In discussion section, figures should be denoted in corresponding discussion paragraph. Otherwise, it would be hard for readers to connect the discussion with the results.
• Figures can be combined if they serve the same arguments. Besides, some figures can be put in supplementary information, such as figure 7 and 10.

Experimental design

• The idea of this paper is that endometrial hypoxia leads to mitochondrial damage and consequently activates PINK1 overexpression. The high-level mitophagy mediated by overexpressed PINK1 upregulates ROS, which increases cell invasion ability and finally contribute to the development of adenomyosis (AM). Although decreased cell invasion ability has been noticed in the knockout experiments of PINK1, the causal relationship between ROS accumulation and cell invasion ability has not been confirmed yet in this study. (Figure 4 confirms the correlation but does not establish causality.) Please provide more evidence for this argument.
• Line 404-406 states that “Higher concentrations can directly induce tumor cell death, while subclinical doses of ROS accumulation below the threshold of tumor cell death, caused by chronic hypoxia, can lead to increased tumor cell invasion ability (Shin et al., 2016b)”. Quantitative analysis of ROS concentration in figure 4 and comparison with literature (“subclinical doses”) would further strengthen this paper.

Validity of the findings

• Some statistical significances are not indicated, such as figure 1B, 6B, 9B and 11B. Please provide all of them. If there is no significance, please indicate ‘n.s.’.

Additional comments

This work studies the role of PINK1 in activating mitophagy and the consequent effects on development of adenomyosis (AM). This is an informative study that combines in vitro cell study, in vivo mouse model and human tissue sample. The paper is written in a professional and clear language. All necessary data, figures, methods are provided. I would suggest minor revision with comments provided.

Cite this review as

Reviewer 2 ·

Basic reporting

- In general, Chen et al present an article with several important deficiencies.
- Except for some minnor errors, professional English is correct.
- Lack of bibliographic references in the introduction and methods.
- The structure and figures of the article are not appropiate enough. Results should be better and more extensively explained. Figures captions include little information and there are some errors in some graphs (ROS probe intensity e.g.). Some results would go directky in supplementary data (KO of PINK1 cotrols).
- Excessive number of figure, there is no point in subdiving many of them.

Experimental design

- The hypothesis of this work would be that hypoxia is behind an increase of ROS, which would promote a mitochondrial dysfunction and engage mitophagy, inducing a progression of AM. However, the experimental design does not include any reference to hypoxic conditions to demonstrate that and link mitochondrial deficiencies with hypoxic-induce oxidative stress.
- Some extra information of the mice model is needed (if you have used this animal previosly, you could explain how they are characterized). Also about the ROS kit, what recognizes this probe?
- It would be advisable to include some more experiments in human tissue (genetic expression, immunohistochemical assays…).

Validity of the findings

- In figure 1, bands of NDP52 western blot are strange (also in the original images of the membranes).
- Figure 4: ROS staining seems to be unspecific. Some extra controls are needed to check this.
- Figure 5: PINK1 staining seems to be unspecific. Some extra controls (2ry antibody e.g.) are needed. Intensity quantification of several images is needed.
- Figure 8: Diagrams do not show what you explain, the colocalization between both lines in control and AM groups are really similar. You should include a Pearson Correlation Coefficient analysis. Moreover, if you want to leave this diagrams, you should include the location of the line that corresponds to each one.
- It would be interesting include some experiments to study mitophagosome formation (colocalization of LC3 with mitochondria, levels of mitophagy-related proteins in isolated mitochondria, etc.). Usually, the accumulation of some autophagy receptors such as p62 appears as a results of some autophagy resolution deficiency. This should be analyzed.
- With these results, authors cannot conclude the link between mitophagy and AM and much less between mitochondrial oxidative stress and increased mitophagy. In fact, authors stablish that a ROS production would increase as a result of induced mitophagy, but mitochondrial autophagy is activated precisely under high mitochondrial damage.

Cite this review as

Reviewer 3 ·

Basic reporting

The research studied the role of PINK1 in the mitophagy process of adenomyosis (AM), suggesting that the high level of PINK1 expression in AM-related disease models comparing to the control group by looking into protein expression levels, so-localization of lysosome and mitochondria, the levels of ROS as well as the invasion level using endometrial stromal cells as a model. In general, the study requires further depth and exploration.

Experimental design

In Figure 1, the protein levels should be normalized to their control group as the control group expression equals 1.0. The figure needs to be re-analyzed.

In figure 4, it seems that the DAPI stain for the AM group was not focused well, please make sure the parameter setting for both groups are the same. Image quality needs to be improved.

The fluorescence data for the co-localization level between mitochondria and lysosome showed very minimum difference for the control and AM group, more convincing data need to be provided in order to justify the author’s hypothesis.

From Figure 9 it seems like protein NDP52 and Parkin, as well as P62 all had a very high expression level comparing to the control group. It would be more convincing if siRNA knockdown was performed on all these proteins individually in order to sort out the importance of each protein.

In Figure 12, for the transwell invasion assay, the difference of the migrated cell per field for the AM condition seems to be very minor comparing to the control group if combining panel B and D. It seems like in panel D the migrated cell per field for the AM condition is way lower then panel B. It would make more sense if the data from two figures are normalized to each other.

Validity of the findings

No comment

Additional comments

The language used in this paper could benefit from improvement, I would suggest that the authors conduct a thorough review of the document and correct for the typographical and/or grammatical errors.

Cite this review as

---

## Round 0.2 · accepted · Accept

All concerns of the reviewers were addressed and the manuscript is acceptable now.

Reviewer 1 ·

Basic reporting

no comment

Experimental design

no comment

Validity of the findings

no comment

Additional comments

All my comments have been addressed. Although the authors did not provide answers to some of the questions I asked, limitations of this work is declared in the revised version. In addition, the revised version provides more explanations and reduces ambiguities.

Cite this review as